# Cytokine Storm Syndrome in SARS-CoV-2 Infections: A Functional Role of Mast Cells

**DOI:** 10.3390/cells10071761

**Published:** 2021-07-12

**Authors:** Bahareh Hafezi, Lily Chan, Jason P. Knapp, Negar Karimi, Kimia Alizadeh, Yeganeh Mehrani, Byram W. Bridle, Khalil Karimi

**Affiliations:** 1Department of Clinical Science, School of Veterinary Medicine, Ferdowsi University of Mashhad, Azadi Square, Mashhad 9177948974, Iran; hafezibahareh@yahoo.com (B.H.); n.karimi@mail.um.ac.ir (N.K.); 2Department of Pathobiology, Ontario Veterinary College, University of Guelph, Guelph, ON N1G 2W1, Canada; lchan12@uoguelph.ca (L.C.); jknapp03@uoguelph.ca (J.P.K.); ymehrani@uoguelph.ca (Y.M.); 3Department of Diagnostic Medicine & Pathobiology, College of Veterinary Medicine, Kansas State University, Manhattan, KS 66506, USA; kimia@vet.k-state.edu

**Keywords:** COVID-19, SARS-CoV-2, mast cells, cytokine storm, vaccine

## Abstract

Cytokine storm syndrome is a cascade of escalated immune responses disposing the immune system to exhaustion, which might ultimately result in organ failure and fatal respiratory distress. Infection with severe acute respiratory syndrome-coronavirus-2 can result in uncontrolled production of cytokines and eventually the development of cytokine storm syndrome. Mast cells may react to viruses in collaboration with other cells and lung autopsy findings from patients that died from the coronavirus disease that emerged in 2019 (COVID-19) showed accumulation of mast cells in the lungs that was thought to be the cause of pulmonary edema, inflammation, and thrombosis. In this review, we present evidence that a cytokine response by mast cells may initiate inappropriate antiviral immune responses and cause the development of cytokine storm syndrome. We also explore the potential of mast cell activators as adjuvants for COVID-19 vaccines and discuss the medications that target the functions of mast cells and could be of value in the treatment of COVID-19. Recognition of the cytokine storm is crucial for proper treatment of patients and preventing the release of mast cell mediators, as impeding the impacts imposed by these mediators could reduce the severity of COVID-19.

## 1. Cytokine Storm Syndrome Occurs during Viral Infection and Inflammation

During an immune response, the cytokine storm phenomenon arises when homeostasis is not returned, and the pro-inflammatory pathways are without regulation and are hyperactive [1,2,3]. Rather than being thought of as a specific disease, the cytokine storm syndrome is considered to be a culminating endpoint to numerous diseases and conditions that occur during an attempt to fight off infections [1,4].

Cytokine storm syndrome was initially observed after systemic infections which took on a similar appearance to that of influenza infections. However, it was difficult to identify since it is not directly associated with any one pathogen or insult to a host, but is rather a common product of the host’s immune system responding to any number of pathogens or insults [1]. There are some commonly observed signs and symptoms in cytokine storm syndrome such as fever, systemic inflammation, multi-organ failure, and high concentrations of circulating cytokines [1,5]. It is crucial to determine the underlying cause of the cytokine storm as it will impact the prognosis, symptoms, and treatments that should be administered [1,2]. An important and difficult distinction that is required to properly treat cytokine storm syndrome is differentiating between the concentrations of cytokines and the magnitude of immune responses needed for fighting off the underlying cause of the cytokine storm, and those that are unnecessary and will ultimately cause harm [1].

Cytokines, including interferons (IFNs), interleukin (IL)-6 and IL-1, are observed to be amplified during immune responses against viruses; all of which are associated with the cytokine storm [2,5]. Multiple viruses, including different subtypes of influenza A virus, have been observed to cause infections that can result in cytokine storm syndrome [3]. The gravity of viral infections is in part due to the degree of virulence as well as the nature of the immune response, which creates an ideal circumstance for the development of cytokine storm syndrome [5]. Although there are general anti-viral responses generated by the body, the overall response produced in the host will differ for each viral infection depending on several factors including the virus’s method of infection, mechanism of action, localization, and viral replication rate; all of which will influence the overall cytokine profiles that will be produced in the host during the immune response, as well as the likelihood of a cytokine storm syndrome developing [2,5].

Treatment for a viral infection that consequently triggers cytokine storm syndrome would be to treat both the infection—with anti-viral medications—and the inflammatory syndrome with anti-inflammatory drugs. A challenge with treating a viral infection that is causing a cytokine storm syndrome is differentiating the dysregulated inflammatory responses from the protective ones [6]. The cytokine profiles induced in the host by the underlying viral infection can assist in determining the most appropriate anti-inflammatory drugs or other therapeutic agents to be administered [3]

## 2. Severe Acute Respiratory Syndrome-Coronavirus (SARS-CoV)-2 May Result in Cytokine Storm Syndrome

Cytokine storm syndrome is a cascade of escalated immune responses that can cause the immune system to become exhausted, which might ultimately result in organ failure and fatal respiratory distress [7]. In the context of infections with SARS-CoV-2 that can cause the coronavirus disease that emerged in 2019 (COVID-19) in some people, cytokine storm syndrome is likely the only feature that contributes to severe cases of the disease [8]. When SARS-CoV-2 infects the body, inflammatory responses play a vital role in the response to the virus. However, unregulated innate and adaptive immune responses of the host caused by SARS-CoV-2 result in uncontrolled production of cytokines and eventually can result in the development of cytokine storm syndrome [9].

Cytokine storm syndrome can lead to apoptosis of epithelial and endothelial cells in the lungs, vascular leakage, and acute respiratory distress syndrome (ARDS) [6]. It is believed that ARDS has been responsible for a substantial number of deaths among patients diagnosed with severe COVID-19. Therefore, ARDS can be considered a characteristic immune-mediated clinical feature of SARS-CoV-2 infections. This is consistent with the existing evidence on the original SARS-CoV and Middle East respiratory syndrome (MERS)-CoV, the other well-known coronavirus infections with considerable resemblance to COVID-19 [10]. Although SARS-CoV-2 has shown phylogenetic similarities to both SARS-CoV and MERS-CoV, it seems more likely to result in a cytokine storm compared to the other two [11].

## 3. Mast Cell (MC) Responses to SARS-CoV-2 May Promote Cytokine Storms 

MCs are granulated leukocytes [12] that reside in mucosal interfaces in close contact with the environment to respond to environmental challenges [13] and infectious organisms [14,15,16]. Recent studies showed that MCs are key effector cells of the innate immune system, and during viral infections they release several inflammatory mediators and cytokines, including histamine and IL-6 [17,18]. Histamine can contribute to the progression of inflammatory responses in many cell types and local tissues by enhancing the secretion of pro-inflammatory cytokines, such as IL-1α, IL-1β, IL-6, and chemokine (C-C motif) ligand 3 (CCL3) [19]. Production of IL-6 might be detrimental during viral infections, promoting virus survival and/or exacerbation of clinical disease [20]. This could be due to polarization of helper T cells into a T helper-2 phenotype (impairing IFNγ production), failure in cytolytic activity, or promoting infected cell survival by inhibiting apoptosis. Knowing that MCs detect multiple classes of viruses, including both DNA and RNA viruses, one may consider MC responses to viruses an enhanced inflammation that has the potential to be harmful [19]. 

Coronaviruses have now become one of the main respiratory pathogens that cause extreme inflammatory outbreaks of acute pneumonia in individuals [21]. Evidence has shown a potential association of MCs with COVID-19, and the activation of MCs located in the submucosa of the respiratory tract by SARS-CoV-2 is known to lead to the release of pro-inflammatory cytokines such as IL-1, IL-6 and TNF-α. Moreover, autopsy findings from the lungs of patients that died from COVID-19 showed accumulation of MCs that was speculated to be the cause of pulmonary edema, inflammation, and thrombosis in COVID-19 pathophysiology [16,22,23].

The primary receptor that the SARS-CoV-2 spike (S) protein uses to facilitate viral entry into cells has been identified as angiotensin-converting enzyme (ACE)-2 [24]. The ACE2 protects against lung injury, and its downregulation is associated with serious lung injuries. Increasing ACE2 concentrations can correct for arterial hypoxia and improve pulmonary circulatory haemodynamics [25]. Interestingly, ACE2 serine protease in MCs is known to transform angiotensin I to angiotensin II [26] and is required for SARS-CoV-2 binding and entry into target cells. Glycoprotein spikes on the virus’s outer envelope bind to the extracellular domain of the ACE2 receptor and allow the virus to enter the target cell [27]. MCs can give rise to bronchoconstriction, both by initiating a renin-angiotensin-generating system in the lungs and by producing leukotrienes [28]. Transmembrane serine protease 2 (TMPRSS2), which can be produced by MCs, is necessary for the priming of the coronavirus spike protein [16], and MC-derived serine protease tryptase, has been observed to be essential for SARS-CoV-2 infection [29,30]. Inhibition of viral entry into the lung cells by serine protease inhibitors such as camostat mesylate has confirmed these findings [31].

Besides pro-inflammatory cytokines and chemokines, MCs are also known to secrete chymase [32], a type of serine protease. Chymase activates transforming growth factor beta (TGF-β), and matrix metalloproteinases [MMP] such as MMP-9 [33] which are involved in pulmonary fibrosis. Moreover, thromboxanes and platelet-activating factor (PAF) produced by MCs can lead to microthrombosis in the lungs and cause COVID-19-associated coagulopathy, which has been confirmed in postmortem analysis of patients who died due to COVID-19 [22].

A study done by Afrin et al. indicated a similarity between the prevalence of MC activation syndrome (MCAS) and that of severe cases of COVID-19. Many aspects of hyper inflammation in patients with COVID-19 seem to be coordinated with MCAS, while drugs which are unlikely to show potency against viral diseases seem to be effective against both MCAS and SARS-CoV-2-induced hyperinflammation [34]. COVID-19 is now widely recognized to be linked to a variety of extra-pulmonary symptoms, such as multisystem inflammatory syndrome (MIS-A). Symptoms of MIS-A and how stress exacerbates them are somewhat similar to those of MCAS [16].

## 4. MC Activators May Enhance COVID-19 Vaccine-Induced Immunity 

Often overshadowed by their involvement in pathology, MCs play an important role in protective immunity as multi-equipped pathogen sensors. MCs express a variety of receptors such as Toll-like receptors (TLRs), FcεRI-III/IgE, and cytosolic sensors that allow them to detect a wide range of pathogens such as bacteria and viruses, as well as toxins and allergens [12]. In the detection of viruses, MCs rely on the expression of TLR2, TLR3, TLR7, TLR8, TLR9, melanoma differentiation-associated protein 5 (MDA5), and retinoic acid-inducible gene (RIG)-I [12]. Activation through these sensors leads to the *de novo* synthesis of cytokines and chemokines for the generation of a virus-focused response as opposed to degranulation mediated through IgE-induced activation [12,35]. In response to viral infection, MCs not only aid in the innate immune response through recruitment of natural killer cells via production of IL-8, but also influence the adaptive response in numerous ways [36]. For example, MC production of TNF-α recruits dendritic cells (DCs) to the site of infection, which then traffic to draining lymph nodes where antigen presentation takes place [37,38]. Interestingly, MCs are able to induce CD8^+^ T cell responses through the modulation of DC phenotype as well as recruit T cells to sites of infection through the production of CCL5 [39]. Due to the ability of MCs to detect a wide array of pathogens and modulate naturally acquired adaptive immunity, many efforts are focused on the identification and characterization of MC activators that may be useful as adjuvants to enhance vaccine-induced immunity. 

In roughly a year since the COVID-19 pandemic began, a handful of vaccines with evidence of protective efficacy, such as BNT162b2 (Pfizer-BioNTech) and mRNA-1273 (Moderna), have been developed and are currently being administered in numerous countries around the world [40,41]. The majority of the COVID-19 vaccines that have been developed target the SARS-CoV-2 S protein with the goal of generating neutralizing Abs capable of binding to SARS-CoV-2 virions to prevent infection [42]. Unfortunately, numerous SARS-CoV-2 variants with mutations in the S protein have appeared and exhibit reduced sensitivity to certain antibodies that could neutralize earlier forms of the virus [43,44,45,46]. Furthermore, many COVID-19 vaccines appear to be less effective against certain SARS-CoV-2 variants. For example, the AstraZeneca COVID-19 vaccine was found to have only 10.4% efficacy against the B.1.351 (501Y.V2) variant that was first identified in South Africa [47]. To further compound issues, slow vaccination programs have introduced a narrow selection pressure against the S protein into the population, which may further promote the emergence of a full SARS-CoV-2 vaccine escape mutant [48,49]. These concerns emphasize the need for second-generation COVID-19 vaccines capable of generating a broader immune response that is not only focused on inducing SARS-CoV-2 S protein-neutralizing antibodies. Second-generation vaccines must also prioritize robust anti-SARS-CoV-2 cellular-mediated immunity through the targeting of additional SARS-CoV-2 proteins rich in T cell epitopes, such as the SARS-CoV-2 nucleocapsid protein [50]. Interestingly, recent studies of individuals who recovered from natural infections suggest that T cell-mediated immunity plays a larger role in protection against SARS-CoV-2 than previously thought [51,52]. Especially important is the T helper cell response induced upon infection. Interestingly, compared with severe cases, people who recovered from mild COVID-19 also had more robust memory CD8^+^ T cell responses in the respiratory tract [50]. Furthermore, long-term immunity and control of re-infection appears to be largely antibody-independent but T cell-dependent [53]. Considering the ability of MCs to respond to viral infections and enhance T cell responses, we promote the investigation of mast cell activators as adjuvants for COVID-19 vaccines with the goal of enhancing the magnitude and longevity of SARS-CoV-2-specific T cell responses.

Here, we summarize two MC activators capable of enhancing anti-viral adaptive immunity that are worthy of further investigation as potential adjuvants in COVID-19 vaccines. A focus is placed on antibody- and cell-mediated responses and the overall T helper cell response.

### 4.1. Compound 48/80 

Compound 48/80 (c48/80) is a widely studied MC activator and is a potent mucosal adjuvant [35]. It is a synthetic polymer that stimulates the degranulation of MCs in an IgE-independent manner to evoke an inflammatory response [54]. Following activation with c48/80, MCs produce TNF-α which promotes DC and T cell functions. leading to significant trafficking of DCs to draining lymph nodes (DLNs) [37]. Numerous studies have demonstrated the ability of c48/80 to induce high concentrations of serum IgG and mucosal soluble IgA when co-administered intranasally with viral glycoproteins [55,56,57]. Additionally, c48/80-adjuvanted responses were shown to enhance protection in challenge models using lethal doses of H1N1 influenza virus, as well as vaccinia virus [56,57]. Interestingly, a follow up study on influenza demonstrated the inability of c48/80 to enhance protection against H1N5 challenge, suggesting that the adjuvanticity of c48/80 may vary depending on the immunogen utilized [58]. Similarly, the T helper cell responses induced by c48/80 also varied between studies [55,57]. One study co-administering c48/80 with the hepatitis B virus glycoprotein found a T helper-2-biased response while the studies focusing on the H1N1 influenza found a balanced T helper-1/T helper-2 response [55,57]. However, it is important to note that the study immunizing with hepatitis B virus glycoprotein utilized a chitosan nanoparticle platform, which may be related to the T helper-2 bias [55]. While the T helper cell response was not characterized in the study utilizing vaccinia virus, c48/80 induced only a modest increase in the number of CD8^+^ T cells, potentially indicating a T helper-2 bias [56]. In contrast with these studies, one group investigated the ability of c48/80 to enhance CD8^+^ T cell-mediated protective immunity utilizing the nucleoprotein of the H1N1 influenza virus [59]. This group found that c48/80 induced high levels of IgG1 and IgG2a in similar proportions, indicating a potentially well balanced T helper-1/T helper-2 response [59]. More importantly, it induced a higher number of nucleoprotein-specific CD8^+^ T cells compared to CD4^+^ T cells. The c48/80+nucleoprotein combination provided 100% protection in mice when challenged with homologous H1N1 influenza virus and even protected against heterologous challenge using the H9N2 strain of influenza virus [59]. While high concentrations of serum IgG and secretory IgA were produced in these mice, the fact that the nucleoprotein is contained within virus particles and infected cells suggests that the nucleoprotein-specific CD8^+^ T cells were the main correlate of protection against virus challenge [59]. Based on these studies, c48/80 appears to be a potent mucosal adjuvant capable of inducing protective antibody- and cell-mediated immunity against certain viruses. While potentially promising for use as an adjuvant in COVID-19 vaccines, the immunogen-dependent effect of c48/80 and variability in T helper cell responses necessitates extensive investigation. 

### 4.2. Interleukin-18

When investigating the adjuvant ability of the IL-1 family of cytokines in combination with a recombinant influenza virus hemagglutinin protein via intranasal vaccination, Kayamuro et al. found that only four cytokines, IL-1α, IL-1β, IL-18, and IL-33 correlated with high concentrations of serum IgG and secretory IgA [60]. Upon further analysis, they discovered that the adjuvancy of IL-18 occurred in a MC-dependent manner and led to the recruitment of DCs and T cells to the site of immunization [60]. Upon re-stimulation of splenocytes *in vitro*, mice treated with IL-18 as an adjuvant had increased concentrations of both T helper-1- and T helper-2-associated cytokines (IFN-y, IL-4, and IL-5), and enhanced numbers of CD8^+^ T cells. In an influenza challenge model, IL-18 significantly enhanced protection (100% survival). Additionally, adjuvanting with IL-18 resulted in induction of a balanced ratio of IgG1/IgG2a. As such, IL-18 may be a useful adjuvant for COVID-19 vaccines.

## 5. COVID-19 Drug Therapies

As mentioned in the previous sections, the role of cytokine storm syndrome has been established in the pathogenesis of COVID-19, and the contribution of MCs to this phenomenon is now established [22]. Activated MCs release proinflammatory cytokines and are mediators of bronchoconstriction resulting in life-threatening inflammatory reactions and pulmonary problems [61]. Therefore, preventing the release of MC mediators or impeding the impacts imposed by these mediators could potentially reduce the severity of COVID-19. Here, we summarize the pharmacologic therapies that have the potential to suppress SARS-CoV-2-induced cytokine storms (Table 1). The medications that target the performance of MCs could be of value in the treatment plans of COVID-19 and require further evaluation. Moreover, the variations in the pharmacodynamics, pharmacokinetics, administration routes, drug interactions, safety, and costs of the medications in each category necessitate more precise assessment and comparison for clinical usage.

### 5.1. TNF-α-Specific Antibodies

TNF-α is released by MC [69] and considered one of the leading mediators of acute inflammation, the activation of which can also trigger IL-1 and IL-6 production [70], both of which can be produced by a variety of cells, including MCs. High concentrations of TNF-α were reported in both plasma and tissues of patients with COVID-19 [71]. Anti-TNF-α medications, namely adalimumab, infliximab, etanercept, certolizumab, and golimumab are effective as immunotherapies in patients with autoimmune inflammation and severe bilateral lung lesions [72,73,74]. The efficacy and acceptable safety of these agents make them promising candidates to be tested for the potential to control cytokine storms in patients with COVID-19 [73]. 

### 5.2. IL-1-Specific Antibodies

MCs secrete IL-1 [75,76] and IL-1 activates MCs to release inflammatory chemical mediators, and cytokines/chemokines [77]. Therefore, IL-1 inhibition could be considered a potential method to dampen cytokine storms and associated injuries in patients with COVID-19. 

#### Anakinra

This agent blocks the binding of IL-1α and IL-1β to IL-1 receptors. it has been evaluated for use in patients with COVID-19 [78]. In an open-label case series study in France, anakinra was administered to nine patients with COVID-19 who were at high risk of worsening disease. In five to eight days, computerized tomography scans of the chests of all patients revealed the cessation of expansion of lesions [63]. A pilot study in Italy reported promising outcomes following early high-dose intravenous administration of anakinra in five moderate-to-severe cases of COVID-19 [64]. Another large retrospective cohort study in Italy examined anakinra in COVID-19 patients with ARDS and hyperinflammation [78]. 

### 5.3. IL-6-Specific Antibodies

MCs produce IL-6 in response to viruses [79] and serum IL-6 has been shown to be a valuable biomarker of COVID-19 [80,81]. Some monoclonal antibodies competitively bind to both membranous and soluble receptors of IL-6, leading to the blockade of the signals mediated by IL-6. 

#### 5.3.1. Tocilizumab

Tocilizumab (Actemra) is a recombinant humanized monoclonal antibody with the potential of inhibiting all membranous or soluble receptors for IL-6 [82]. This agent has been taken into consideration for cytokine release syndrome [83] and retrospective studies in COVID-19 patients demonstrated that tocilizumab might be clinically beneficial [65,66]. Clinical trials on the safety and efficacy of tocilizumab have been performed (NCT04335071, ChiCTR2000029765, and NCT04317092) and the medication has been recommended in protocols, such as the COVID-19 care handbook by the Italian Society of Infectious and Tropical Diseases [84]. 

#### 5.3.2. Siltuximab 

This is a chimeric monoclonal antibody that blocks IL-6 by binding to it and studies suggested that this medication holds potential for the treatment of COVID-19 [85,86]. 

#### 5.3.3. Clazakizumab

This agent is a monoclonal engineered humanized antibody that binds directly to IL-6 with high affinity [87]. It is being tested in the United States in a phase 2 study in patients with COVID-19 (NCT04348500).

#### 5.3.4. Sarilumab

This fully human monoclonal antibody under the generic name Kevzara, can bind to both soluble and membrane-bound IL-6 receptors and was first approved for treatment of rheumatoid arthritis [88]. It is under evaluation for efficacy and safety in a double-blind, phase 2/3 clinical trial of hospitalized patients with severe COVID-19 (NCT04315298).

#### 5.3.5. Sirukumab

This medicine prevents the binding of IL-6 to membrane receptors [89]. Therefore, it can hinder the biological effects of this cytokine and could be examined for the control of cytokine storms caused by SARS-CoV-2.

### 5.4. Janus Kinase (JAK) Inhibitors

JAK3 [90,91], a member of the Janus family protein-tyrosine kinases, is expressed in MCs and the JAK1/JAK2-inhibitor ruxolitinib inhibits MC degranulation and cytokine release [92]. Therefore, another potential approach for suppressing cytokine storms is to target the JAK-STAT pathway and interrupt the signaling pathways of cytokines. The advantage of these medications over cytokine-specific antibodies is the inhibition of several cytokines at once [93].

#### 5.4.1. Baricitinib

Baricitinib is a selective inhibitor of JAK1 and JAK2 and believed to interfere with the signaling of key cytokines that are produced abundantly in COVID-19, including IL-2, IL-6, IL-7, and granulocyte monocyte colony-stimulating factor [94]. It has also been suggested to be capable of inhibiting viral entry into cells [95]. An open-label study in Italy (NCT04358614) indicated a significant reduction in the concentration of C-reactive protein, improved oxygen saturation, and reduced fever [67]. Several other clinical trials are being initiated to evaluate baricitinib in patients with COVID-19.

#### 5.4.2. Ruxolitinib

Ruxolitinib under the generic name Jakafi is also a selective inhibitor of JAK1 and JAK2 [73,74]. Some clinical trials are examining this drug in cases of COVID-19. Fedratinib is a highly selective inhibitor of JAK2 and a modest inhibitor of JAK1 and JAK3 [93] showed efficacy against cytokine storms caused by COVID-19 via the downregulation of T helper-17-associated cytokines [68]. Other JAK inhibitors that could be taken into consideration for COVID-19 include upadacitinib, oclacitinib, peficitinib, and tofacitinib, which inhibit various JAKs or combinations thereof [74]. However, two concerns remain about the administration of anti-JAK drugs; the blockade of IFN-α, which is an essential cytokine against viral infections, and the risk of thrombocytopenia and/or thrombosis.

### 5.5. Targeting MCs

There are a variety of drugs that target functional aspects of MC biology. These are shown in Table 2 and could be a useful set of therapeutic agents in the clinical toolbox to control cytokine storms that occur in patients diagnosed with COVID-19.

#### 5.5.1. MC Stabilizers

This group of medications block the degranulation of MCs and prevent the secretion of histamine and other mediators, leading to diminished cytokine release by diverse cell types [93]. MC stabilizers are categorized as synthetic, semi-synthetic, and plant-derived [125] and have the potential to dampen cytokine storms in cases of COVID-19.

##### Cromolyn Sodium

Cromolyn sodium was originally introduced to treat allergic asthma and has quickly shown to be effective for treating disorders related to MCs such as mastocytosis, allergic rhinitis, conjunctivitis, and intestinal allergies. Cromolyn sodium has been shown to act as a G-protein-coupled receptor 35 (GPR35) agonist that is expressed in human MCs. Interestingly, GPR35 mRNA is upregulated upon challenge with IgE antibodies [96], suggesting that GPR35 may be a potential target for using cromolyn sodium in the treatment of asthma. Previous studies on the pharmacological actions of cromolyn sodium indicate that it prevents the release of mediators from mast cells [126], although recent studies employing mouse [127] and human [128] cultured MCs questioned cromolyn sodium function as a MC stabilizer agent. Regardless, the mechanism of action for this agent has been noted as inhibiting the activity of protein kinase C and transport of chloride into MCs [97]. Furthermore, cromolyn blocks the activity of macrophages, the release of eicosanoids and cytokines, and the expression of adhesion molecules [98]. It was reported that in a mouse model of influenza, this agent regulated the expression of IL-6, TNF-α, TLR3, and TRIF, leading to the improvement of injuries and inflammatory reactions in the nose, trachea, and lungs [99]. Cromolyn sodium was also shown to potently inhibit Nsp12, which is a conserved coronavirus protein acting as an RNA-dependent RNA polymerase in viral replication and transcription [100]. Furthermore, this medication was found to exert a direct impact on chloride and calcium ion channels in coronaviruses interfering with their biologic cycle [93,129]. It should be taken into consideration that cromolyn sodium is currently deemed to be effective in decreasing inflammation and cytokine storms in patients with COVID-19 [130,131].

##### Ketotifen

As another MC stabilizer, ketotifen exerts an antagonistic effect against the histamine-1 receptor [101]. It was suggested to decrease lung lesions and apoptosis during the early stages of infection with H5N1 influenza virus [102].

##### Flavonoids

These are a large group of natural stabilizers, including luteolin, disometin, apigenin, kaempferol, fisetin, quercetin, morin, genistein, and epigallocatechin gallate [111,112,113]. Luteolin was reported to suppress the release of histamine, leukotrienes, prostaglandin 2, and granulocyte/monocyte colony-stimulating factor *in vitro* from human MCs [104]. Moreover, it hindered the production of TNF-α and IL-6 in murine MCs [103]. Examinations revealed that luteolin targeted virus entry into the host cells and could significantly hinder SARS-CoV activity *in vitro* [105]. Fisetin and quercetin diminished the expression of TNF-α, IL-1β, IL-6, and IL-8 by MCs [106,112]. It was observed that quercetin and its derivatives were able to constrain SARS-CoV serine proteases, which are of importance in the infectivity of the virus [107]. Quercetin also downregulated the transcription of histidine decarboxylase in MCs, as an enzyme involved in histamine generation [108]. Genistein was indicated to dose-dependently inhibit the degranulation of MCs and phosphorylation of some cellular proteins that contribute to the signaling of activated MCs [109]. Epigallocatechin gallate, found in green tea, showed anti-allergic effects by decreasing degranulation and histamine release in rat MCs [110].

##### Coumarins

Coumarins are another group of herbal agents that stabilize MCs. Scopletin, scaporone, and artekeiskeanol A decreased the secretion of TNF-α and IL-6 [113]. Also, artekeiskeanol A and selinidin had the potential to suppress the phosphorylation of some molecules involved in signaling events in MCs [114,115].

##### Terpenoids

Terpenoid inhibits antigen-IgE-induced degranulation. Some terpenoids, namely parthenolide and dehydroleucodine have been noted as influential MC stabilizers during the recent decades [113,116].

##### Alkaloids

Sinomenine and theanine were demonstrated to inhibit MC degranulation induced by antigens, diminish the synthesis of IL-4, IL-1, IL-6, and TNF-α, as well as preventing the phosphorylation of signaling molecules [113]. Furthermore, theanine is suggested to stabilize MCs by preventing the perturbation of the lipid bilayers of these cells [117]. Xestospongin C prevents the increase in intracellular calcium ion levels, which is essential for MC exocytosis [118].

##### Palmitoylethanolamide

The analgesic and anti-inflammatory actions of this molecule are attributed to the inhibitory influence on the release of histamine and TNF-α from MCs [119]. Moreover, adelmidrol, as a palmitoylethanolamide analogue, was capable of reducing MCs degranulation in a rat model of chronic inflammation [132]. A systemic review of randomized clinical trials on the usage of this medication for the common cold and influenza demonstrated protective potential against viral infections of the respiratory tract [133].

##### Antihistamines

Anti-inflammatory and MC-stabilizing effects have been shown for some antihistamine medicines, namely olopatadine [120,121,134], rupatadine [16,122,135], and ketotifen [136,137], which inhibit release of histamine or other MC mediators. Therefore, these antihistaminic MC stabilisers and mediator blockers could be considered in the treatment of SARS-CoV-2 infection. It was concluded in a review of 18 randomized clinical trials that antihistamines exert limited short-term positive effects on the severity of common cold symptoms [138].

##### Miscellaneous

The inhibitors of tyrosine protein kinases, namely compound-13, R-112, and ER-27317 are among the synthetic stabilizers of MCs, which can prevent signal transduction in allergic reactions [113]. Furthermore, hypothemycin was shown to block the FcεRI-mediated activation of MCs and cytokine release through constraining kit kinase [139]. *Thymoquinone* is a phytochemical compound, which was indicated to inhibit degranulation in meningeal MCs [140]. In a study, C_70_-tetraglycolic acid [TGA], a water-soluble fullerence, dampened the degranulation of human skin MCs [141]. Orazipone [OR-1384] and its derivative OR-1958 are recent sulphydryl anti-inflammatory agents that can reduce TNF-α production by human MCs [142].

#### 5.5.2. Inhibitors of MC Mediators

##### Leukotriene Mediators

Leukotrienes are molecules secreted from MCs, basophils, and eosinophils contribute remarkably to asthma pathogenesis. Cysteinyl leukotrienes (CysLTs) induce the contraction of smooth muscles, swelling, and edema in the airways. Leukotriene B4 acts as a chemoattractant and activates neutrophils and eosinophils [93]. Both synthesis and binding of leukotrienes to their receptors could be targeted. The synthesis inhibitors, such as zileuton, target the 5-lipoxygenase enzyme. The other group, including telukast, zafirlukast, and montelukast, blocks the CysLT1 receptor on the cells that CysLTs target [123]. It has been noted that montelukast may also have an antiviral impact through targeting SARS-Cov-2 3CL protease, which is one of the key proteases in coronaviruses [100].

##### MC Proteases

MCs produce different proteases, such as tryptase. Camostat mesylate, an inhibitor of tryptase, was observed to prevent the entry of SARS-CoV into cells by targeting TMPRSS2, leading to the blockade of virus spread and pathogenesis [143]. Also, SARS-CoV relies on TMPRSS2 for the priming of S proteins, which is needed for virus entry into cells lining the respiratory tract [124].

## 6. Conclusions

In some people, SARS-CoV-2 can cause the uncontrolled production of cytokines and the development of cytokine storm syndrome. Evidence indicates that MCs can respond to SARS-CoV-2 and accumulate in the lungs of patients with COVID-19, where they correlate with pulmonary edema, inflammation, and thrombosis. MCs are foundational drivers of inflammation [144] that produce preformed inflammatory mediators and cytokines upon activation. Preventing the release of mast cell-derived mediators and impeding the impacts imposed by these mediators could blunt the severity of COVID-19. Additionally, MC activators could be considered for testing as adjuvants for COVID-19 vaccines. Further, the medications that target the performance of MCs could be potentially of value in the treatment of COVID-19. The recognition of the cytokine storm initiated by MCs is crucial for the proper treatment of COVID-19 in patients and could potentially lead to novel clinical approaches for many pathological conditions in which cytokine storm or cytokine release syndromes are life-threatening features.

## Figures and Tables

**Table 1 cells-10-01761-t001:** Anti-cytokines and Janus kinase (JAK)-inhibitors under study for COVID-19.

Medication	Mechanism of Action	References and Clinical Trials
Infliximab	Binds to soluble and transmembrane forms of TNF-α	NCT04425538, NCT04734678, NCT04593940, NCT04344249
Etanercept	TNF-α antagonist	[62]
Anakinra	Blocks the binding of IL-1α and IL-1β to IL-1 receptors	[63,64], NCT04443881, NCT04680949, NCT04366232, NCT04412291, NCT04364009, NCT04324021, NCT04339712, NCT04362111, NCT04643678, NCT04330638, NCT04341584, NCT04381936, NCT02735707
Tocilizumab	Blocks IL-6 receptors	[65,66], ChiCTR2000029765, NCT04445272, NCT04479358, NCT04317092, NCT04412772, NCT04331795, NCT04332094, NCT04377659, NCT04730323, NCT04346355, NCT04409262, NCT04372186, NCT04332913, NCT04320615, NCT04600141, NCT04363736, NCT04779047, NCT04435717, NCT04377750, NCT04577534, NCT04412291, NCT04335071, NCT04678739, NCT04356937, NCT04363853, NCT04335305, NCT04310228, NCT04403685, NCT04560205, NCT04519385, NCT04339712, NCT04322773, NCT04476979
Siltuximab	Prevents binding of IL-6 to both soluble and membrane-bound IL-6 receptors	NCT04329650, NCT04322188, NCT04486521
Clazakizumab	Binds to IL-6	NCT04494724, NCT04343989, NCT04348500, NCT04363502, NCT04659772
Sarilumab	Blocks IL-6 receptors	NCT04315298, NCT04661527, NCT04357808, NCT04386239, NCT04341870, NCT04359901, NCT04327388
Sirukumab	Binds to IL-6	NCT04380961
Baricitinib	Inhibits JAK1 and JAK2	[67], NCT04421027, NCT04693026, NCT04373044, NCT04346147, NCT04401579, NCT04390464, NCT04640168, NCT04321993
Ruxolitinib	Inhibits JAK1 and JAK2	NCT04355793, NCT04377620, NCT04334044, NCT04366232, NCT04362137, NCT04338958, NCT04477993, NCT04359290, NCT04581954, NCT04348695, NCT04403243
Fedratinib	Inhibits JAK2 (JAK2-selective inhibitor with higher inhibitory activity for JAK2 over family members JAK1, JAK3 and TYK2)	[68]
Tofacitinib	Inhibits all JAKs	NCT04415151, NCT04750317, NCT04469114

**Table 2 cells-10-01761-t002:** Potential medications for COVID-19-associated cytokine storms through targeting of MCs.

Medication	Mechanism of Action	References and Clinical Trials
**Mast cell stabilizers**		
Cromolyn sodium	- Known as a MC stabilizer (based on rat MC studies) and act as a G-protein-coupled receptor 35 (GPR35) agonist - Inhibits protein kinase C and chloride transport into MCs - Blocks macrophages, the release of eicosanoids and cytokines, and the expression of adhesion molecules - Regulates the expression of IL-6, TNF-α, TLR3, and TRIF - Inhibits Nsp12 - Directly affects chloride and calcium ion channels	[93,96,97,98,99,100]
Ketotifen	- Antagonizes the histamine-1 receptor - Decreases lung lesions and apoptosis in the early stages of H5N1 infection	[101,102]
Flavonoids	- Suppresses the release of histamine, leukotrienes, PGF2, and GM-CSF - Targets virus entry into the host cells - Reduces the expression of TNF-α, IL-1β, IL-6, and IL-8 by MC - Constrains SARS-CoV serine proteases - Downregulates histidine decarboxylase in MCs - Inhibits MC degranulation and phosphorylation of some cell signaling proteins	[103,104,105,106,107,108,109,110,111,112] NCT04680819, NCT04468139, NCT04487964, NCT04622865
Coumarins	- Decreases TNF-α and IL-6 secretion - Suppresses the phosphorylation of some MC signaling molecules	[113,114,115]
Terpenoids	- Inhibits antigen-IgE-induced degranulation	[116] NCT04487964
Alkaloids	- Inhibits MC degranulation. - Diminishes the synthesis of IL-4, IL-1, IL-6, and TNF-α. - Prevents phosphorylation of signaling molecules - Hinders the perturbation of the lipid bilayers of MCs - Prevents the increase in intracellular calcium ion levels, which is essential for exocytosis by MCs	[113,117,118] NCT04479202
Palmitoylethanolamide	- Inhibits release of histamine and TNF-α by MCs	[119] NCT04568876, NCT04619706
Antihistamines	- Anti-inflammatory and MC-stabilizing effects have been shown for some antihistamine medicines	[120,121,122], NCT04370262, NCT04389567, NCT04504240, NCT04724720, NCT04565392, NCT04545008
Inhibitors of MC mediators		
Zileuton	- Reduces the synthesis of leukotrienes through targeting 5-lipoxygenase enzyme	
Telukast, Zafirlukast, and Montelukast	- Inhibits the binding of leukotrienes through blocking the CysLT1 receptor - Montelukast targets SARS-Cov-2 3CL protease	[100,123] NCT04714515, NCT04718285
Proteases	- Prevents virus entry into cells through targeting the transmembrane protease serine 2	[124]

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
