# Peer review of "Cytokine Storm Syndrome in SARS-CoV-2 Infections: A Functional Role of Mast Cells"

_cells, 2021, doi:10.3390/cells10071761_

Round 1
Reviewer 1 Report
The manuscript by Hafezi et al has been written in a manner that indicates the role of mast cells upon the COVID-19 infection. The authors have covered how mast cells function as inducing the development of cytokine storm syndrome and described how mast cell suppression can be associated with COVID-19 therapeutics.
This manuscript is informative and is generally written well with rigor. The text is presented clearly, and the logical flow justifies the authors' points.
No point needs to be addressed.
Author Response
We would like to thank the reviewers for their careful assessment of our manuscript and constructive comments. We have made changes based on reviewers’ suggestions and we feel this has improved the manuscript.
Reviewer 1. The manuscript by Hafezi et al has been written in a manner that indicates the role of mast cells upon the COVID-19 infection. The authors have covered how mast cells function as inducing the development of cytokine storm syndrome and described how mast cell suppression can be associated with COVID-19 therapeutics.
This manuscript is informative and is generally written well with rigor. The text is presented clearly, and the logical flow justifies the authors' points.
No point needs to be addressed.
Response: We would like to thank the reviewer for the careful assessment of our manuscript. We appreciate the comment.
Reviewer 2 Report
This is a fairly well organized review that very weak when it comes to mast cells and related inteventions
Major points
Key papers about mast cells releasing cytokines are missing
Taracanova A, Tsilioni I, Conti P, Norwitz ER, Leeman SE, Theoharides TC. Substance P and IL-33 administered together stimulate a marked secretion of IL-1β from human mast cells, inhibited by methoxyluteolin. Proc Natl Acad Sci U S A. 2018 Oct 2;115(40):E9381-E9390. doi: 10.1073/pnas.1810133115. Epub 2018 Sep 19. PMID: 30232261; PMCID: PMC6176605.
Taracanova A, Alevizos M, Karagkouni A, Weng Z, Norwitz E, Conti P, Leeman SE, Theoharides TC. SP and IL-33 together markedly enhance TNF synthesis and secretion from human mast cells mediated by the interaction of their receptors. Proc Natl Acad Sci U S A. 2017 May 16;114(20):E4002-E4009. doi: 10.1073/pnas.1524845114. Epub 2017 May 1. PMID: 28461492; PMCID: PMC5441798.
Patel AB, Theoharides TC. Methoxyluteolin Inhibits Neuropeptide-stimulated Proinflammatory Mediator Release via mTOR Activation from Human Mast Cells. J Pharmacol Exp Ther. 2017 Jun;361(3):462-471. doi: 10.1124/jpet.117.240564. Epub 2017 Apr 12. PMID: 28404689.
The section on cromolyn is problematic as the mechanisms listed have not been documented in human mast cells and does not include many papers indicating it does not inhibit mast cells
Oka T, Kalesnikoff J, Starkl P, Tsai M, Galli SJ. Evidence questioning cromolyn's effectiveness and selectivity as a 'mast cell stabilizer' in mice. Lab Invest. 2012 Oct;92(10):1472-82. doi: 10.1038/labinvest.2012.116. Epub 2012 Aug 20. PMID: 22906983; PMCID: PMC3580174.
Weng Z, Zhang B, Asadi S, Sismanopoulos N, Butcher A, Fu X, Katsarou-Katsari A, Antoniou C, Theoharides TC. Quercetin is more effective than cromolyn in blocking human mast cell cytokine release and inhibits contact dermatitis and photosensitivity in humans. PLoS One. 2012;7(3):e33805. doi: 10.1371/journal.pone.0033805. Epub 2012 Mar 28. PMID: 22470478; PMCID: PMC3314669.
The section on flavonoids is missing key papers and reviews
Kandere-Grzybowska K, Kempuraj D, Cao J, Cetrulo CL, Theoharides TC. Regulation of IL-1-induced selective IL-6 release from human mast cells and inhibition by quercetin. Br J Pharmacol. 2006 May;148(2):208-15. doi: 10.1038/sj.bjp.0706695. PMID: 16532021; PMCID: PMC1617055.
Weng Z, Patel AB, Panagiotidou S, Theoharides TC. The novel flavone tetramethoxyluteolin is a potent inhibitor of human mast cells. J Allergy Clin Immunol. 2015 Apr;135(4):1044-1052.e5. doi: 10.1016/j.jaci.2014.10.032. Epub 2014 Dec 10. PMID: 25498791; PMCID: PMC4388775.
The section on alkaloids is problematic as many alkaloids also stimulate mast cells
The section on antihistamines is wrong-they do NOT inhibit mast cell degranulation, except maybe for rupatadine
Vasiadi M, Kalogeromitros D, Kempuraj D, Clemons A, Zhang B, Chliva C, Makris M, Wolfberg A, House M, Theoharides TC. Rupatadine inhibits proinflammatory mediator secretion from human mast cells triggered by different stimuli. Int Arch Allergy Immunol. 2010;151(1):38-45. doi: 10.1159/000232569. Epub 2009 Aug 6. PMID: 19672095; PMCID: PMC7065400.
Author Response
We would like to thank the reviewers for their careful assessment of our manuscript and constructive comments. We have made changes based on reviewers’ suggestions and we feel this has improved the manuscript.
Reviewer 2. This is a fairly well organized review that very weak when it comes to mast cells and related inteventions
Response: We would like to thank the reviewer for the constructive comments. We have adopted nearly all your suggestions and feel this has improved the manuscript.
Major points
Key papers about mast cells releasing cytokines are missing
Taracanova A, Tsilioni I, Conti P, Norwitz ER, Leeman SE, Theoharides TC. Substance P and IL-33 administered together stimulate a marked secretion of IL-1β from human mast cells, inhibited by methoxyluteolin. Proc Natl Acad Sci U S A. 2018 Oct 2;115(40):E9381-E9390. doi: 10.1073/pnas.1810133115. Epub 2018 Sep 19. PMID: 30232261; PMCID: PMC6176605.
Taracanova A, Alevizos M, Karagkouni A, Weng Z, Norwitz E, Conti P, Leeman SE, Theoharides TC. SP and IL-33 together markedly enhance TNF synthesis and secretion from human mast cells mediated by the interaction of their receptors. Proc Natl Acad Sci U S A. 2017 May 16;114(20):E4002-E4009. doi: 10.1073/pnas.1524845114. Epub 2017 May 1. PMID: 28461492; PMCID: PMC5441798.
Patel AB, Theoharides TC. Methoxyluteolin Inhibits Neuropeptide-stimulated Proinflammatory Mediator Release via mTOR Activation from Human Mast Cells. J Pharmacol Exp Ther. 2017 Jun;361(3):462-471. doi: 10.1124/jpet.117.240564. Epub 2017 Apr 12. PMID: 28404689.
Response: We have incorporated the publications into the manuscript body.
The section on cromolyn is problematic as the mechanisms listed have not been documented in human mast cells and does not include many papers indicating it does not inhibit mast cells
Oka T, Kalesnikoff J, Starkl P, Tsai M, Galli SJ. Evidence questioning cromolyn's effectiveness and selectivity as a 'mast cell stabilizer' in mice. Lab Invest. 2012 Oct;92(10):1472-82. doi: 10.1038/labinvest.2012.116. Epub 2012 Aug 20. PMID: 22906983; PMCID: PMC3580174.
Weng Z, Zhang B, Asadi S, Sismanopoulos N, Butcher A, Fu X, Katsarou-Katsari A, Antoniou C, Theoharides TC. Quercetin is more effective than cromolyn in blocking human mast cell cytokine release and inhibits contact dermatitis and photosensitivity in humans. PLoS One. 2012;7(3):e33805. doi: 10.1371/journal.pone.0033805. Epub 2012 Mar 28. PMID: 22470478; PMCID: PMC3314669.
Response: We have refined table 2 and revised the related paragraph as follow:
Cromolyn sodium was originally introduced to treat allergic asthma and has quickly shown to be effective for treating disorders related to MCs such as mastocytosis, allergic rhinitis, conjunctivitis, and intestinal allergies. Cromolyn sodium has been shown to act as a G-protein-coupled receptor 35 (GPR35) agonist that is expressed in human MCs. Interestingly, GPR35 mRNA is upregulated upon challenge with IgE antibodies (Yang et al., 2010) suggesting that GPR35 may be a potential target for using cromolyn sodium in the treatment of asthma. Previous studies on the pharmacological actions of cromolyn sodium indicate that it prevents the release of mediators from mast cells (Rodríguez et al., 2020), although recent studies employing mouse (Oka et al., 2012) and human (Weng et al., 2012) cultured MCs questioned cromolyn sodium function as a MC stabilizer agent. Regardless, the mechanism of action for this agent has been noted as inhibiting the activity of protein kinase C and transport of chloride into MCs (Storms and Kaliner, 2005). Furthermore, cromolyn blocks the activity of macrophages, the release of eicosanoids and cytokines, and the expression of adhesion molecules (Sinniah et al., 2017). It was reported that in a mouse model of influenza, this agent regulated the expression of IL-6, TNF-α, TLR3, and TRIF, leading to the improvement of injuries and inflammatory reactions in the nose, trachea, and lungs (Han et al., 2016). Cromolyn sodium was also shown to potently inhibit Nsp12, which is a conserved coronavirus protein acting as an RNA-dependent RNA polymerase in viral replication and transcription (Wu et al., 2020). Furthermore, this medication was found to exert a direct impact on chloride and calcium ion channels in coronaviruses interfering with their biologic cycle (Abdin et al., 2020; Alton and Norris, 1996). It should be taken into consideration that cromolyn sodium is currently deemed to be effective in decreasing inflammation and cytokine storms in patients with COVID-19 (Malone et al., 2021; Yousefi et al., 2021).
The section on flavonoids is missing key papers and reviews
Kandere-Grzybowska K, Kempuraj D, Cao J, Cetrulo CL, Theoharides TC. Regulation of IL-1-induced selective IL-6 release from human mast cells and inhibition by quercetin. Br J Pharmacol. 2006 May;148(2):208-15. doi: 10.1038/sj.bjp.0706695. PMID: 16532021; PMCID: PMC1617055.
Weng Z, Patel AB, Panagiotidou S, Theoharides TC. The novel flavone tetramethoxyluteolin is a potent inhibitor of human mast cells. J Allergy Clin Immunol. 2015 Apr;135(4):1044-1052.e5. doi: 10.1016/j.jaci.2014.10.032. Epub 2014 Dec 10. PMID: 25498791; PMCID: PMC4388775.
Response: The citations for the section was revised, accordingly.
The section on alkaloids is problematic as many alkaloids also stimulate mast cells
Response: We thank the reviewer for the comment but since the compounds that we had named in the section work as MC stabilizers, we kept the section as it was.
The section on antihistamines is wrong-they do NOT inhibit mast cell degranulation, except maybe for rupatadine
Vasiadi M, Kalogeromitros D, Kempuraj D, Clemons A, Zhang B, Chliva C, Makris M, Wolfberg A, House M, Theoharides TC. Rupatadine inhibits proinflammatory mediator secretion from human mast cells triggered by different stimuli. Int Arch Allergy Immunol. 2010;151(1):38-45. doi: 10.1159/000232569. Epub 2009 Aug 6. PMID: 19672095; PMCID: PMC7065400.
Response: We appreciate the reviewer’s constructive comment. We have refined table 2 and revised the related paragraph as follows:
Anti-inflammatory and MC-stabilizing effects have been shown for some antihistamine medicines, namely olopatadine (Baba et al., 2015; Leonardi and Quintieri, 2010; Rosenwasser et al., 2005), rupatadine (Alevizos et al., 2013; Theoharides and Conti, 2020b; Vasiadi et al., 2010), and ketotifen (Okayama and Church, 1992; Sokol et al., 2013), which inhibit release of histamine or other MC mediators. Therefore, these antihistaminic MC stabilisers and mediator blockers could be considered in the treatment of SARS-CoV-2 infection. It was concluded in a review of 18 randomized clinical trials that antihistamines exert limited short-term positive effects on the severity of common cold symptoms (De Sutter et al., 2015).
Reviewer 3 Report
This is a review article about Cytokine Storm (CS) and its relationship with Mast Cells (MC). Overall, the article is well written, leading the reader to several primary concepts about the topic, culminating in describing the mechanism of action of several drugs that have already been proposed in the treatment of CS, including CM modulators.
However, I believe the article is too long with many unnecessary paragraphs containing concepts, I believe, already well known to current "Cells" readers.
I would indeed advise the authors to considerably summarize chapter 1 (1. Cytokine Storm Syndrome Occurs during Viral Infection and Inflammation) from line 32 to line 121, leaving only the concepts strictly necessary to understand the article.
It seems that the main objective of this review would be the role of MC in the pathogenesis of CS induced by SARS-CoV-2. If so, chapter 1 would be too long and unnecessary.
Considering the title of this review - Cytokine Storm Syndrome in SARS-CoV-2 Infections: A Functional Role of Mast Cells - I would start the article from chapter 2 and introduce in this chapter 2 some of the strictly necessary concepts that were covered in chapter 1.
Likewise, I would either delete chapter 5 completely from 5.1 to 5.10 or summarize this part of the paper, building a very robust table 1, containing information such as the drug name, more details about the mechanism of action, and more details and results of the main trials. As the central theme of the review seems to be the role of MC in COVID-19 supporting possible treatments involving the mechanism of action of these cells, it seems that chapter 5 (5.1-5.10) deals with various forms of treatment of SC, is very long. I would focus on drugs with actions on MCs (from item 5.11) and table 2, focusing on the main objective of the review.
Finally, lines 182-183 (...and MC-derived serine protease tryptase, has been observed to be essential for SARS-CoV-2 infection (Schwartz, 1994)).... I do not think this reference is correct since it is 1994. I suggest reviewing.
Author Response
We would like to thank the reviewers for their careful assessment of our manuscript and constructive comments. We have made changes based on reviewers’ suggestions and we feel this has improved the manuscript.
Reviewer 3. This is a review article about Cytokine Storm (CS) and its relationship with Mast Cells (MC). Overall, the article is well written, leading the reader to several primary concepts about the topic, culminating in describing the mechanism of action of several drugs that have already been proposed in the treatment of CS, including CM modulators. However, I believe the article is too long with many unnecessary paragraphs containing concepts, I believe, already well known to current "Cells" readers. I would indeed advise the authors to considerably summarize chapter 1 (1. Cytokine Storm Syndrome Occurs during Viral Infection and Inflammation) from line 32 to line 121, leaving only the concepts strictly necessary to understand the article.
It seems that the main objective of this review would be the role of MC in the pathogenesis of CS induced by SARS-CoV-2. If so, chapter 1 would be too long and unnecessary.
Considering the title of this review - Cytokine Storm Syndrome in SARS-CoV-2 Infections: A Functional Role of Mast Cells - I would start the article from chapter 2 and introduce in this chapter 2 some of the strictly necessary concepts that were covered in chapter 1.
Response: We sincerely thank the reviewer for the meticulous assessment of our manuscript and constructive comments. We have shortened the section 1 (from 9 paragraphs to 4 short paragraphs) of the manuscript based on reviewers’ suggestions and we feel this has improved the manuscript substantially.
Likewise, I would either delete chapter 5 completely from 5.1 to 5.10 or summarize this part of the paper, building a very robust table 1, containing information such as the drug name, more details about the mechanism of action, and more details and results of the main trials. As the central theme of the review seems to be the role of MC in COVID-19 supporting possible treatments involving the mechanism of action of these cells, it seems that chapter 5 (5.1-5.10) deals with various forms of treatment of SC, is very long. I would focus on drugs with actions on MCs (from item 5.11) and table 2, focusing on the main objective of the review.
Response: We again thank the reviewer for the thoughtful and constructive comments. We have shortened section 5 (5.1 through 5.10) of the manuscript to be more concise while maintaining the information required to understand table 1.
Finally, lines 182-183 (...and MC-derived serine protease tryptase, has been observed to be essential for SARS-CoV-2 infection (Schwartz, 1994)).... I do not think this reference is correct since it is 1994. I suggest reviewing.
Response: We apologize for the mistake, and we have changed the citation, accordingly.
“MC-derived serine protease tryptase, has been observed to be essential for SARS-CoV-2 infection(Gebremeskel et al., 2021; Gioia et al., 2020).”
Round 2
Reviewer 3 Report
Accept in present form